# Non-Canonical G-quadruplexes cause the hCEB1 minisatellite instability in *Saccharomyces cerevisiae*

Aurèle Piazza[1†‡], Xiaojie Cui[1†], Michael Adrian[2], Frédéric Samazan[1], Brahim Heddi[2§], Anh-Tuan Phan[2*], Alain G Nicolas[1*]

[1]Institut Curie, CNRS UMR3244, PSL Research University, Paris, France; [2]School of Physical and Mathematical Sciences, Nanyang Technological University, Singapore

**\*For correspondence:** phantuan@ntu.edu.sg (A-TP); alain.nicolas@curie.fr (AGN)

[†]These authors contributed equally to this work

**Present address:** [‡]Department of Microbiology and Molecular Genetics, University of California, Davis, Davis, United States; [§]Laboratoire de Biologie et de Pharmacologie Appliquée, ENS Paris-Saclay, Université Paris Saclay, Cachan Cedex, France

**Competing interests:** The authors declare that no competing interests exist.

**Abstract** G-quadruplexes (G4) are polymorphic four-stranded structures formed by certain G-rich nucleic acids in vitro, but the sequence and structural features dictating their formation and function in vivo remains uncertain. Here we report a structure-function analysis of the complex hCEB1 G4-forming sequence. We isolated four G4 conformations in vitro, all of which bear unusual structural features: *Form 1* bears a V-shaped loop and a snapback guanine; *Form 2* contains a terminal G-triad; *Form 3* bears a zero-nucleotide loop; and *Form 4* is a zero-nucleotide loop monomer or an interlocked dimer. In vivo, *Form 1* and *Form 2* differently account for 2/3[rd] of the genomic instability of hCEB1 in two G4-stabilizing conditions. *Form 3* and an unidentified form contribute to the remaining instability, while *Form 4* has no detectable effect. This work underscores the structural polymorphisms originated from a single highly G-rich sequence and demonstrates the existence of non-canonical G4s in cells, thus broadening the definition of G4-forming sequences.

## Introduction

G-rich nucleic acids can form G-quadruplexes (G4), a stable four-stranded structure formed by stacking of guanine tetrads (G-quartets) in the presence of coordinating cations such as $K^+$ (*Figure 1A*) (*Davis, 2004*; *Neidle, 2009*; *Patel et al., 2007*; *Sen and Gilbert, 1988*). This core tetrad organization is the signature of a G4, around which a variety of conformations blossom depending on the primary sequence and physico-chemical conditions (*Chen and Yang, 2012*). Furthermore, competitive structural polymorphisms can result from a single nucleic acid sequence, when multiple contiguous G-tracts are available (*Phan, 2010*). These complexities challenge our ability to predict G4 formation from any particular sequence, let alone predict a particular structure.

Based on pioneering biophysical knowledge, a G4 consensus motif of the form $G_{3-5}N_{1-7} G_{3-5}N_{1-7} G_{3-5}N_{1-7}G_{3-5}$ (where N can be any nucleotide) was adopted (*Huppert and Balasubramanian, 2005*; *Todd et al., 2005*). It imposed constraints on the G-tract number (4) and length (3 to 5 nt) as well as on the length of each connecting loop (1 to 7 nt) (*Figure 1A*). These parameters established a reasonable compromise balancing false-positive (containing sequences with several loops of >4 nt [*Guédin et al., 2010*; *Rachwal et al., 2007*]) and false-negative motifs such as G4s containing only two G-quartets (*Macaya et al., 1993*; *Chinnapen and Sen, 2004*) or a single long loop together with two other short loops (*Guédin et al., 2010*). This consensus was extensively used to mine genomic sequences, and estimated ~376,000 potential G4-forming motifs in the human genome (*Huppert and Balasubramanian, 2005*; *Todd et al., 2005*). However, recent structural studies unveiled additional 'non-canonical' G4, bearing bulges (*De Nicola et al., 2016*; *Mukundan and Phan, 2013*), strand interruptions with snapback guanines (*Adrian et al., 2014*), and incomplete

**eLife digest** Molecules of DNA encode the information needed to build cells and keep them alive. DNA is made of two strands that contain several different chemical groups known as bases arranged in different orders, like letters and words in a phrase. Generally, two DNA strands wrap around each other to make a three dimensional structure known as a double helix. However, in certain circumstances, some sequences of DNA bases can adopt alternative structures. For example, DNA sequences that contain lots of a base known as guanine may sometimes form structures called G-quadruplexes in which sets of four guanines come together.

G-quadruplexes are involved in many processes in cells including regulating the activity of genes, but they can also interfere with the process that replicates the DNA at each generation. This causes the cell's genetic information to be modified, which can damage the cell and can promote cancer. However, it is difficult to predict which DNA sequences are susceptible to form G-quadruplexes and what consequence their folding might have on the biological processes happening in cells.

Recent computational and biophysical studies have shown that G-quadruplexes can form a larger variety of structures than previously known. Piazza et al. studied how some of these new "non-canonical" structures form in yeast cells and how they may interfere with DNA copying. The experiments show that a single guanine-rich DNA sequence can form several types of non-canonical G-quadruplex structures in yeast cells. This includes structures that do not have complete sets of guanines at their center or are missing loops that connect the bases to one another. Further experiments demonstrate that the threat posed by these G-quadruplexes is linked to the length of their connecting loops and how well their three-dimensional structures withstand heat.

The findings of Piazza et al. identify a set of DNA sequences that are capable of forming G-quadruplexes that harm the cell. The next challenge will be to develop specific molecules that can stabilize the structures of G-quadruplexes. In the future, this avenue of research may aid the development of new treatments for cancer that target specific DNA structures.

tetrads (G-triad) (*Heddi et al., 2016*; *Li et al., 2015*). They result from sequences lacking four G-triplets, and thus escape the consensus. Recently, a high throughput in vitro polymerase stop assay performed on purified human genomic DNA in the presence of $K^+$ or G4-stabilizing ligand Pyridostatin identified 716,310 G4-forming sites; 451,646 sites did not match the consensus (*Chambers et al., 2015*), indicating that the false-negative rate of the initial consensus is massive. Accordingly, a new G4 prediction algorithm (G4Hunter) emphasizing G-richness and skewness over well-defined G-tracts and arbitrary loop lengths has been developed and its predictability (95%) established upon biophysical characterization of hundreds of sequences over an extensive range of thermal stabilities (*Bedrat et al., 2016*). This algorithm conservatively heightened the figure for putative G4 sequences in the human genome to ~700,000, in agreement with the G4-seq assay (*Chambers et al., 2015*). This re-evaluation has implications for inference of *cis*-acting functions of G4 and their association with other genomic and epigenomic features. Hence, biological evidence for the relevance of these non-canonical G4s is paramount.

Compelling evidence for the role of G4s in various biological processes have accumulated (for reviews see: [*León-Ortiz et al., 2014*; *Maizels and Gray, 2013*; *Rhodes and Lipps, 2015*; *Tarsounas and Tijsterman, 2013*; *Vasquez and Wang, 2013*]). Yet, an uncertainty remains between the ability of a predicted sequence to form a G4 in vitro and exert a G4-dependent biological function in vivo. Relevant to the present study, we recently showed that, among a set of validated G4-forming variant sequences of the human minisatellite CEB25, only the G4s with short loops preferentially containing pyrimidine were capable of inducing genomic instability in the eukaryotic model organism *S. cerevisiae* (*Piazza et al., 2015*). These results demonstrated that only a subset of G4-forming sequences actually formed and/or exerted a biological effect; in this case, the ability to interfere with leading strand DNA replication (*Lopes et al., 2011*).

While the unstable CEB25-G4 motif variant bearing short loops matched the G4 consensus (*Piazza et al., 2015*), we also previously reported that the human minisatellite CEB1 was similarly unstable despite the lack of a consensus G4 motif (*Lopes et al., 2011*; *Piazza et al., 2010*,

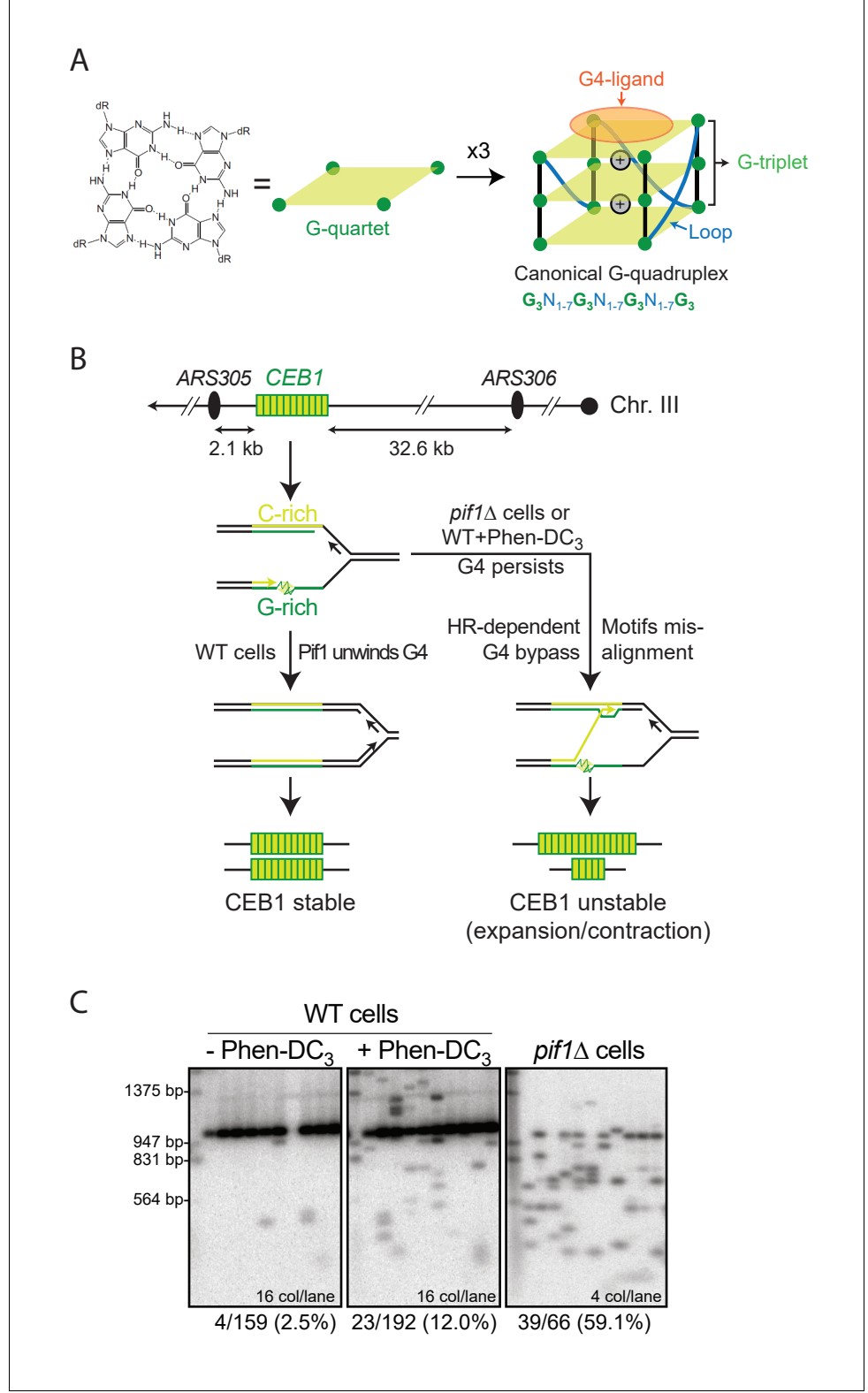

**Figure 1.** G-quadruplexes and G4-dependent minisatellite instability in *S.cerevisiae*. (**A**) Schematic representation of the overall G4 structure, its features and the underlying canonical G4 motif. N can be any nucleotide. G4-ligands such as Phen-DC$_3$ bind by stacking on an outermost G-quartet. (**B**) Site of CEB1 integration in the yeast genome, near the replication origin *ARS305*. CEB1 is oriented so that the G-rich strand is template for the leading strand replication machinery emanating from *ARS305*. Model for G4-dependent minisatellite instability during

*Figure 1 continued*

leading-strand replication (*Lopes et al., 2011*). (C) Example of CEB1 instability in untreated or Phen-DC$_3$-treated WT cells and in a *pif1Δ* mutant. The main band above the 947 bp marker is the parental size CEB1. The Southern blots were published previously in (*Piazza et al., 2015*).

---

*2012*; *Ribeyre et al., 2009*). Our first biophysical study suggested that the CEB1 motif was forming a mixture of several G4 conformations in solution that could not be individually resolved (*Ribeyre et al., 2009*). The structural analyses of an isolated conformation revealed a rather unique snapback scaffold with single-nucleotide loops (*Form 1*, see below and [*Adrian et al., 2014*]), which resulted from a non-consensus G4 motif involving a G-doublet. To demonstrate the in vivo relevance of this unusual form and further resolve the variety of G4s that CEB1 likely forms, we now report our comprehensive biophysical, structural and biological structure-function analysis of the wild-type CEB1 motif and 27 mutated variants, assayed for their effects on genomic instability. This study demonstrates the existence of at least three types of non-canonical G4s in vivo and the threat they pose to genomic stability.

## Results

### Experimental system

Our experimental system assays the instability of G4-prone tandem repeats (expressed as contraction or expansion of the number of motifs) in two G4-stabilizing conditions: upon deletion of the G4-unwinding helicase Pif1 (*Paeschke et al., 2013*; *Ribeyre et al., 2009*) or in cells treated with the G4-stabilizing ligand Phen-DC$_3$ (*De Cian et al., 2007*; *Monchaud et al., 2008*), which inhibits G4 unwinding by Pif1 in vitro and in vivo (*Piazza et al., 2010*). We summarize in *Figure 1B* the mechanism of G4-dependent rearrangement formation during leading strand replication. The CEB1 motif (39 nt) is 77% GC-rich with a GC-skew of 77%. It comprises seven G-tracts: one G-sextet (G$_{9-14}$), three G-triplets (G$_{2-4}$, G$_{16-18}$ and G$_{20-22}$) and three G-doublets (G$_{24-25}$, G$_{31-32}$, G$_{34-35}$) (*Figure 2A*). We previously showed that the CEB1 instability depends on its ability to form G4(s) in vivo by simultaneously mutating the G-sextet and each G triplet (CEB1-Gmut in *Table 1*) (*Ribeyre et al., 2009*). Here, to precisely elucidate the sequences required to form G4(s) and trigger CEB1 instability, we synthesized 26 new minisatellites of similar length bearing single or multiple mutations in each motif of the array (*Table 1*). All constructs were inserted in the vicinity of the *ARS305* origin of replication, in the orientation where the G-rich strand is the template for leading strand synthesis (*Figure 1B*) (*Lopes et al., 2011*). The rearrangement frequencies were measured upon mitotic growth of untreated and Phen-DC$_3$-treated wild-type yeast cells (WT) as well as in *pif1Δ* cells as previously described (*Lopes et al., 2011*; *Piazza et al., 2010*) (example given *Figure 1C*, Materials and methods). The sequences, rearrangement frequencies and statistical comparisons are reported in *Table 1*. First, we measured the rearrangement frequencies of the control CEB1-WT-20 and CEB1-WT-25 alleles (20 and 25 motifs, respectively). These synthetic alleles were stable in WT cells (0% and 2.5% instability, respectively) and significantly unstable both in Phen-DC$_3$-treated WT cells (10.9% and 12% instability, respectively) and in the *pif1Δ* mutant (25.6% and 59.1% instability, respectively) (*Table 1* and *Figure 2B*). The rearrangement frequencies for CEB1-WT-25 were reported previously (*Figure 1C*) (*Piazza et al., 2015*).

### CEB1 instability relies on two G-triplets and short loop length

To address the involvement of individual G-tracts in CEB1 instability, we synthesized mutated minisatellites 19–26 motifs-long and compared their instability to the CEB1-WT allele bearing the closest number of motifs (*Table 1*). First, we assessed the involvement of the G$_{2-4}$, G$_{16-18}$, and G$_{20-22}$ G-triplets by substituting the central guanine by a thymine. Clearly, the CEB1-G3T variant exhibited a significant instability upon Phen-DC$_3$ treatment and *PIF1* deletion (8.9% and 20.0%) (*Table 1* and *Figure 2B*). Compared to the CEB1-WT of a similar size, G3T instability in each assay was not significantly different. In contrast, the CEB1-G17T and CEB1-G21T alleles remained stable in both assays

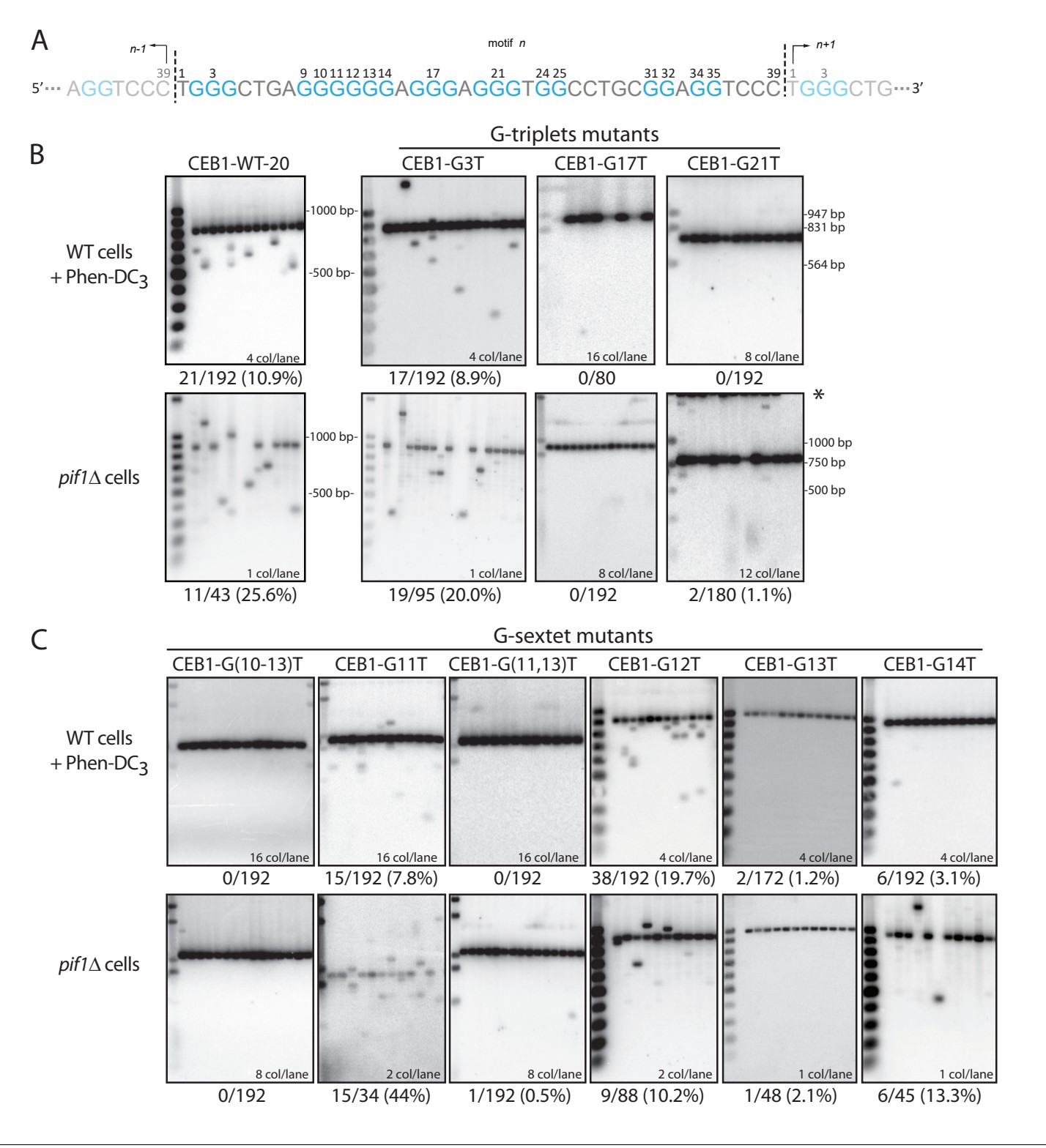

**Figure 2.** Determination of the G-tract requirements for CEB1 instability in *S.cerevisiae*. (**A**) CEB1 motif with guanine tracts colored in cyan and numbered. (**B–C**) Southern blot of CEB1-WT-20 and CEB1 variant alleles mutated for their G-tracts in WT cells treated with Phen-DC$_3$ or in *pif1Δ* cells. (**B**) G-triplet and (**C**) G-sextet mutants. In most instances, several independent colonies were pooled and extracted. The number of colonies analyzed per lane and the total rearrangement frequency is indicated for each blot. The fragment sizes (bp) of the molecular ladders run in the first lane of each blot are indicated.

*Figure 2 continued on next page*

*Figure 2 continued*

The following figure supplement is available for figure 2:

**Figure supplement 1.** Effect of loop length and sequence on CEB1 instability.

($\leq$1.1% rearrangements), significantly different from CEB1-WT (*Table 1* and *Figure 2B*). Thus, the $G_{16-18}$ and $G_{20-22}$ triplets are critical for CEB1 instability while $G_{2-4}$ is dispensable (see below).

To assay the role of loop length, we mutated the $A_{19}$ loop (located between two essential G-triplets) into the more deleterious T (*Piazza et al., 2015*). As expected, this single nucleotide substitution significantly increased the genomic instability of CEB1 in the Phen-DC$_3$-treated WT cells (from 10.9% to 29.7%) and in the *pif1Δ* strain (from 25.6% to 50.0%) (*Figure 2—figure supplement 1*). Furthermore, extending the size of this loop in CEB1-A19TT and CEB1-A19TTT gradually reduced the minisatellite instability to low or background levels (*Figure 2—figure supplement 1*). Hence, we confirmed with the complex CEB1 sequence that G4s bearing short ($\leq$3 nt) pyrimidine-containing loops are more prone to trigger genomic instability.

## Dissection of the G-sextet

Next, we addressed the contribution of the $G_{9-14}$ sextet by performing various combinations of G-to-T substitutions. These mutations, which interrupt G-tracts and simultaneously increase loop length(s), resulted in varying degrees of stabilization. Most dramatically, the G(9-10,14)T, G(10-13)T, G(11,13)T and G13T constructs clearly abolished CEB1 instability in both conditions (*Table 1* and *Figure 2C*). The CEB1-G(11-12)T allele, only assayed in Phen-DC$_3$-treated WT cells, also remained stable (*Table 1*). In contrast, the G(9-10)T, G(9-11)T and G11T mutations had no effect in both conditions (*Table 1* and *Figure 2C*). The CEB1-G14T allele exhibited an intermediate instability, significantly lower than CEB1-WT in both conditions, yet not abolished (*Table 1* and *Figure 2C*). Finally, G12T was unusual since it exhibited a significantly higher level of instability than CEB1-WT in Phen-DC$_3$-treated WT cells (19.6 vs. 12.0%, p-value=0.05) and a lower instability in the absence of Pif1 (10.2 vs. 59.1%, p-value<0.01) (*Table 1* and *Figure 2C*). These results show that not all the Gs in the G-sextet contribute to CEB1 instability.

With the exception of the G12T substitution, which is discussed below, all the mutations that disrupt the $G_{12-14}$ triplet either strongly reduced or abolished CEB1 instability. In contrast, mutations of $G_{9-11}$ triplets had either no or modest effects on CEB1 instability. Hence, critical for CEB1 instability is the presence of a G-triplet in the G-sextet, immediately contiguous with the two other essential $G_{16-18}$ and $G_{20-22}$ triplets. Shifting this $G_{12-14}$ triplet by a single nucleotide away from $G_{20-22}$ (in CEB1-G(9-10,14)T and CEB1-G14T) significantly reduced or abolished CEB1 instability, consistent with the stabilizing effect of increased loop length. In conclusion, our mutational analysis identified three G-triplets ($G_{12-14}$, $G_{16-18}$, and $G_{20-22}$) separated by single nucleotide loops as necessary for sustaining most of the CEB1 instability. Since the distant fourth $G_{2-4}$ triplet is dispensable for CEB1 instability, the sequence causing the bulk of CEB1 instability escapes the $G_3N_{1-7}G_3N_{1-7}G_3N_{1-7}G_3$ consensus. We further sought to determine the origin of the remaining guanines involved in CEB1 instability.

## CEB1 instability largely relies on a G-doublet

To address the involvement of the additional G tracts, we generated mutant alleles of the $G_{24-25}$, $G_{31-32}$ and $G_{34-35}$ doublets, keeping the rest of the CEB1 motif intact. Strikingly, the CEB1-G(24-25)T and CEB1-G25T alleles exhibited a significant ~3 fold decrease of instability compared to CEB1-WT upon Phen-DC$_3$ treatment (3.6% and 4.2% compared to 12%) and *PIF1* deletion (9.4% and 16.1% compared to 59.1%) (*Table 1* and *Figure 3A*). In contrast, for both conditions, the CEB1-G(31-32)T and CEB1-G(34-35)T mutants were as unstable as the similarly sized CEB1-WT arrays (*Table 1* and *Figure 3B*). These results demonstrate the importance of the $G_{24-25}$ doublet in CEB1 instability, contributing to ~2/3$^{rd}$ of the CEB1-WT instability. The unexpected role of this G-doublet is consistent with the additional genetic, biophysical and structural data reported below.

**Table 1.** Genomic instabilities of CEB1 variants and the associated G4 thermal stability.

| Allele | Motifs | Sequence | WT cells | WT cells + Phen-DC3 | pif1Δ cells | Forms | G4 TmUV (°C) |
|---|---|---|---|---|---|---|---|
| CEB1-1.8 (*Lopes et al., 2011*) | 42 | | 0.5% (384) | 11.2% (992) | 56.3% (119) | all | 71.16 (1.78) |
| WT-25 (*Piazza et al., 2015*) | 25 | TGGGCTGAGGGGGGGAGGGGGGGAGGGGGTGGCCTGCGCGGAGGTCCC | 2.5% (159) | 12.0% (192)* | 59.1% (66)* | | |
| WT-20 | 20 | | 0 (96) | 10.9% (192)* | 25.6% (43)* | | |
| WT-10 | 10 | | 0 (96) | 4.2% (176) | ND | | |
| **Single G-tracts mutations** | | | | | | | |
| Gmut (*Ribeyre et al., 2009*) | 20 | TGCGCTGAGCGCGGAGTGAGAGTGGCCTGCGGGAGGTCCC | 1.0% (96) | 1.0% (192) | ND | - | NA |
| | 42 | TGCGCTGAGCGCGGAGTGAGAGTGGCCTGCGGGAGGTCCC | ND | ND | 0 (384) | | |
| G3T | 22 | TGTGCTGAGGGGGGGAGGGGGGGTGGCCTGCGCGGAGGTCCC | 1.0% (96) | 8.9% (192)* | 20.0% (95)* | 1, 2, 3 | 59.74 (0.74) |
| G(9,10)T | 24 | TGGGCTGATTGGGAGGGAGGGTGGCCTGCGGAGGTCCC | 0 (92) | 8.3% (192)* | ND | 1, 2 | 65.8 (0.95) |
| G(9-11)T | 24 | TGGGCTGATTTGGGAGGGAGGGTGGCCTGCGGAGGTCCC | 0 (96) | 11.2% (188)* | 34.4% (64)*† | 2 | 48.81 (2.65) |
| G(9,10,14)T | 21 | TGGGCTGATTGGGTAGGGAGGGTGGCCTGCGGAGGTCCC | 0 (192) | 0 (192)† | 0.5% (192)† | (2) | 44.74 (0.31) |
| G(10-13)T | 25 | TGGGCTGAGTTTTGAGGGAGGGGAGGGTGGCCTGCGGGAGGTCCC | 0 (176) | 0 (192)† | 0 (192)† | - | 46.24 (1.08) |
| G11T | 25 | TGGGCTGAGGTGGGAGGGAGGGTGGCCTGCGGGAGGGTCCC | 0.5% (192) | 7.8% (192)* | 44.4% (36)* | 1, 2 | 70.18 (2.00) |
| G(11-12)T | 24 | TGGGCTGAGGTTGAGGGAGGGTGGCCTGCGGGAGGGTCCC | 0 (96) | 0.5% (192) | ND | - | ND |
| G(11,13)T | 26 | TGGGCTGAGGTGTGAGGGAGGGTGGCCTGCGGGAGGTCCC | 0 (192) | 0 (192)† | 0.5% (192)† | - | 44.02 (0.78) |
| G12T | 24 | TGGGCTGAGGGGTGAGGGAGGGTGGCCTGCGGGAGGTCCC | 0 (92) | 19.7% (192)*† | 10.2% (88)*† | - | 41.11 (0.81) |
| G13T | 24 | TGGGCTGAGGGGGTGAGGGAGGGTGGCCTGCGGGAGGTCCC | 0 (96) | 1.2% (172)† | 2.1% (48)† | - | 42.92 (0.98) |
| G14T | 20 | TGGGCTGAGGGGGGTAGGGAGGGTGGCCTGCGGGAGGTCCC | 0 (96) | 3.1% (192)† | 13.3% (45)*† | (1, 2) | 63.44 (0.92) |
| G17T | 24 | TGGGCTGAGGGGGGAGTGAGGGTGGCCTGCGGGAGGTCCC | 0 (128) | 0 (80)† | 0 (192)† | - | 44.97 (0.52) |
| G21T | 19 | TGGGCTGAGGGGGGAGGGAGTGTGGCCTGCGGGAGGTCCC | 0.5% (180) | 0 (192)† | 1.0% (180)† | 4 | 46.18 (1.14) |
| G(24-25)T | 26 | TGGGCTGAGGGGGGAGGGAGGGTTTCCTGCGGGAGGTCCC | 0 (192) | 3.6% (192)*† | 9.4% (96)*† | 3, 4 | 46.68 (1.21) |
| G25T | 26 | TGGGCTGAGGGGGGAGGGAGGGTGTCCTGCGGGAGGTCCC | 0 (192) | 4.2% (192)*† | 16.1% (87)*† | 3, 4 | 48.91 (0.71) |
| G(31-32)T | 24 | TGGGCTGAGGGGGGAGGGAGGGTGGCCTGCTTAGGTCCC | 1.0% (96) | 13.5% (192)* | 43.2% (37)* | All | 69.03 (2.68) |
| G(34-35)T | 21 | TGGGCTGAGGGGGGAGGGAGGGTGGCCTGCGGATTCCC | 5.0% (96) | 14.4% (180)* | 17.6% (85)* | All | 69.16 (0.22) |
| **Double G-tracts mutations** | | | | | | | |
| G(3,9-11)T | 24 | TGTGCTGATTTGGGAGGGAGGGTGGCCTGCGGGAGGTCCC | 0 (92) | 6.3% (192)* | 8.3% (48)† | 2 | 42.20 (0.33) |
| G(3,24-25)T | 22 | TGTGCTGAGGGGGGAGGGAGGGTTTCCTGCGGGAGGTCCC | 0 (96) | 3.7% (188)† | 1.1% (93)† | 3 | 35.38 (0.29) |
| G(9-10,24-25)T | 20 | TGGGCTGATTGGGAGGGAGGGTTTCCTGCGGGAGGTCCC | 0 (96) | 1.5% (192)† | ND | - | 43.92 (0.65) |
| G(9-11,25)T | 24 | TGGGCTGATTTGGGAGGGAGGGTGTCCTGCGGGAGGTCCC | 0 (96) | 0.5% (192)† | 0 (48)† | - | 42.32 (0.37) |
| G(12,25)T | 24 | TGGGCTGAGGGGTGGAGGGAGGGTGTCCTGCGGGAGGTCCC | 0 (96) | 17.7% (192)* | 11.5% (96)*† | - | ND |
| G(14,24-25)T | 20 | TGGGCTGAGGGGGGTAGGGAGGGTTTCCTGCGGGAGGTCCC | 0 (96) | 0 (188)† | ND | - | 42.10 (0.49) |

* p-values vs. untreated WT < 0.05.

† p-value vs. CEB1-WT < 0.05.

Forms in parenthesis denote a loop modification.

Numbers in parenthesis indicate total number of colonies tested.

In the $T_m$ column, the number in parenthesis indicates the standard deviation of the triplicate $T_m$ measurement.

ND: Not determined; NA: Not applicable

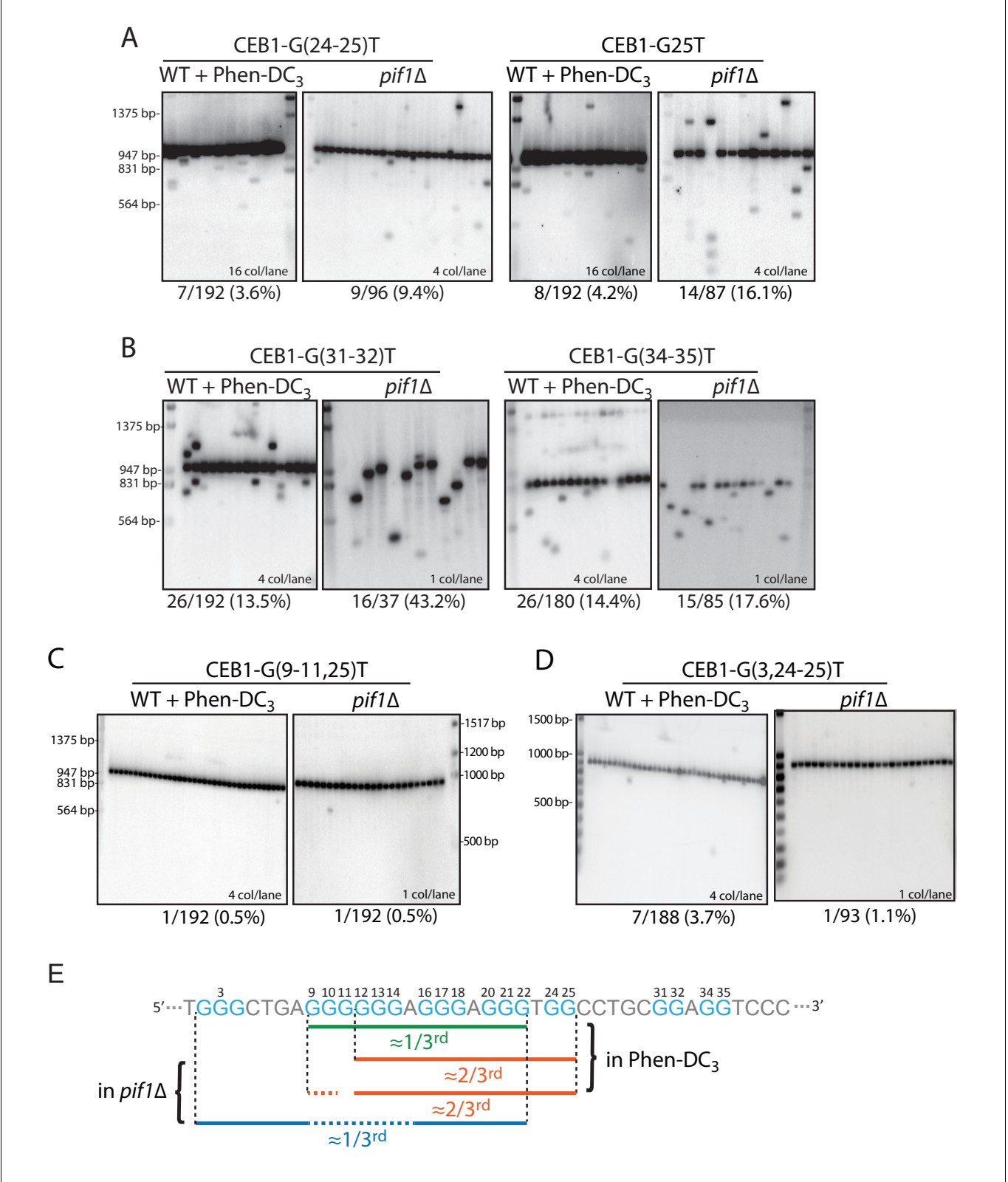

**Figure 3.** Role of the CEB1 $G_{24-25}$-doublet in CEB1 instability. (**A**) Mutation of the $G_{24-25}$ doublet in CEB1-G25T or CEB1-G(24-25)T reduced by $2/3^{rd}$ the minisatellite instability in both *pif1Δ* (ORT7172 and ORT7173) and Phen-DC₃-treated WT cells (ORT7164 and ORT7165). (**B**) Mutation of the $G_{31-32}$ and $G_{34-35}$ doublets did not affect CEB1 instability in *pif1Δ* cells (ANT1965 and ANT1967) and Phen-DC₃-treated WT cells (ANT1950 and ANT1952). (**C**) Mutation of the G-sextet in CEB1-G(9-11)T-G25T abolished the $G_{24-25}$-independent instability in both WT Phen-DC₃-treated (ANT1973) and *pif1Δ*

*Figure 3 continued on next page*

*Figure 3 continued*

(ANT2604) cells. (**D**) Mutation of the dispensable $G_{2-4}$ triplet in CEB1-G(3,24–25)T did not decrease the $G_{24-25}$-doublet-independent instability in Phen-DC₃-treated WT cells (ANT2620) but abolished it in a *pif1Δ* strain (ANT2627). (**E**) Mapping of the overlapping sets of G-tracts required for full CEB1 instability in the WT Phen-DC₃-treated or *pif1Δ* cells.

## Different G-tract requirements in the Phen-DC₃ and the *pif1* contexts

We then investigated the G-tracts requirements for the remainder of $G_{24-25}$-independent instability by generating double-tract mutants. Specifically, we combined the G(24-25)T mutations with the G-sextet G(9-10)T, G(9-11)T or G14T mutations which by themselves had no detectable or only partial effects. In all cases, the remaining $G_{24-25}$-independent instability was consistently abolished ($\leq 1.5\%$) in the Phen-DC₃-treated cells (*Table 1* and *Figure 3C*). Accordingly, in the absence of Pif1, the instability was also abolished in the G(9-11, 25)T construct (<0.5%), significantly less than the 16.1% observed in the single G25T mutant (*Table 1*). These results indicate that the remaining $G_{24-25}$-independent CEB1 instability requires the G-sextet in both conditions. To further assay a potential interaction with the distant $G_{2-4}$ triplet which by itself had no detectable effect, we also constructed the double-tract CEB1-G(3,24–25)T allele. In Phen-DC₃-treated WT cells, it behaved like CEB1-G(24-25)T (3.7% vs. 3.6%) (*Table 1* and *Figure 3D*). Differently, in Pif1-deficient cells this additional G3T mutation ablated the remaining $G_{24-25}$-independent instability (1.1% vs. 9.4% or 16.1% in G(24-25)T and G25T, respectively, p-values<0.05). These data indicate that in the absence of Pif1, but not in in the WT+Phen-DC₃ context, the $G_{24-25}$-independent instability uniquely relies on the distant $G_{2-4}$ triplet. Altogether, these results suggest that multiple G4s contribute to CEB1 instability, and these G4s are not necessarily the same in the presence of the Phen-DC₃ ligand or in the absence of the Pif1 helicase (see Discussion).

In summary, our mutational analyses identified three partially overlapping sets of G-tracts required for full instability, two of which do not match the G4 consensus (*Figure 3E*). One ($G_3AG_3AG_3TG_2$) accounts for ~2/3$^{rd}$ of the instability in both G4-stabilizing conditions: it uniquely involves the $G_{24-25}$-doublet. Another ($G_6AG_3AG_3$) accounts for the remaining ~1/3$^{rd}$ of the instability in the Phen-DC₃ context only: it uniquely involves the entire $G_{9-14}$-sextet and lacks a third non-null loop. The last one ($G_3N_4G_6AG_3AG_3$) accounts for the remaining ~1/3$^{rd}$ of the instability in the *pif1Δ* context only: it uniquely involves the distant $G_{2-4}$ triplet separated from the rest of the G-tracts by 4 nts.

## The CEB1 minisatellite sequence forms multiple G4s

Mapping of the G-tracts required for CEB1 genomic instability suggested the participation of at least three non-canonical G4s. This prompted us to conduct the biophysical characterization of the multiple G4 conformations by NMR spectroscopy. Imino proton spectra of the CEB1 motif (*39-nt*, *Figure 4A,B*) or its G-rich segment (*25-nt*, *Figure 4—figure supplement 1*) evidently indicated the formation of multiple G4s, but no well-defined conformation could be identified. Consequently, we truncated the sequence to represent 4 combinations of G tracts, the structural characteristics of which were assessed using NMR and circular dichroism (CD) spectroscopy. Each of the sequence exhibited a clear NMR spectra corresponding to intramolecular G4s, named *Forms 1* to *4* (*Figure 4B*). Moreover, the G-to-T mutation of the first G-tract (in *25-nt*-[G3T] and *39-nt-shift*-[G3T]) or those on the last two G-tracts (in *25-nt*-G(21,24–25)T) reduced the conformational multiplicity or potential aggregation of the *25-nt* sequence and generated an imino proton spectrum resembling that of *Form 1* or *Form 4*, respectively (*Figure 4—figure supplement 1*). CD spectra of these forms exhibited a positive peak at 260 nm and a negative peak at 240 nm, characteristic of a parallel-stranded G4 (*Figure 4C* and *Figure 4—figure supplement 2*) (*Wieland and Hartig, 2007*). A schematic representation of the four G4 structures isolated from the CEB1 motif is shown in *Figure 4D*. Interestingly, all these conformations exhibited non-canonical structural features. *Form 1* was previously shown to be a parallel, single-nucleotide loop G4 bearing an interrupted strand and a snapback guanine with a V-shaped loop. *Form 1* can exist as a monomer and dimerize at high concentration (*Adrian et al., 2014*). *Form 2* can be a G4 resembling *Form 1* but lacks the $G_9$ or $G_{10}$ snapback guanine, leaving a vacant guanine in the lowermost quartet. Such G-triad-bearing G4s have recently been reported by us and others (*Heddi et al., 2016*; *Li et al., 2015*). *Form 3* that incorporates only a G-sextet and two G-triplets separated by a single residue can fold into a

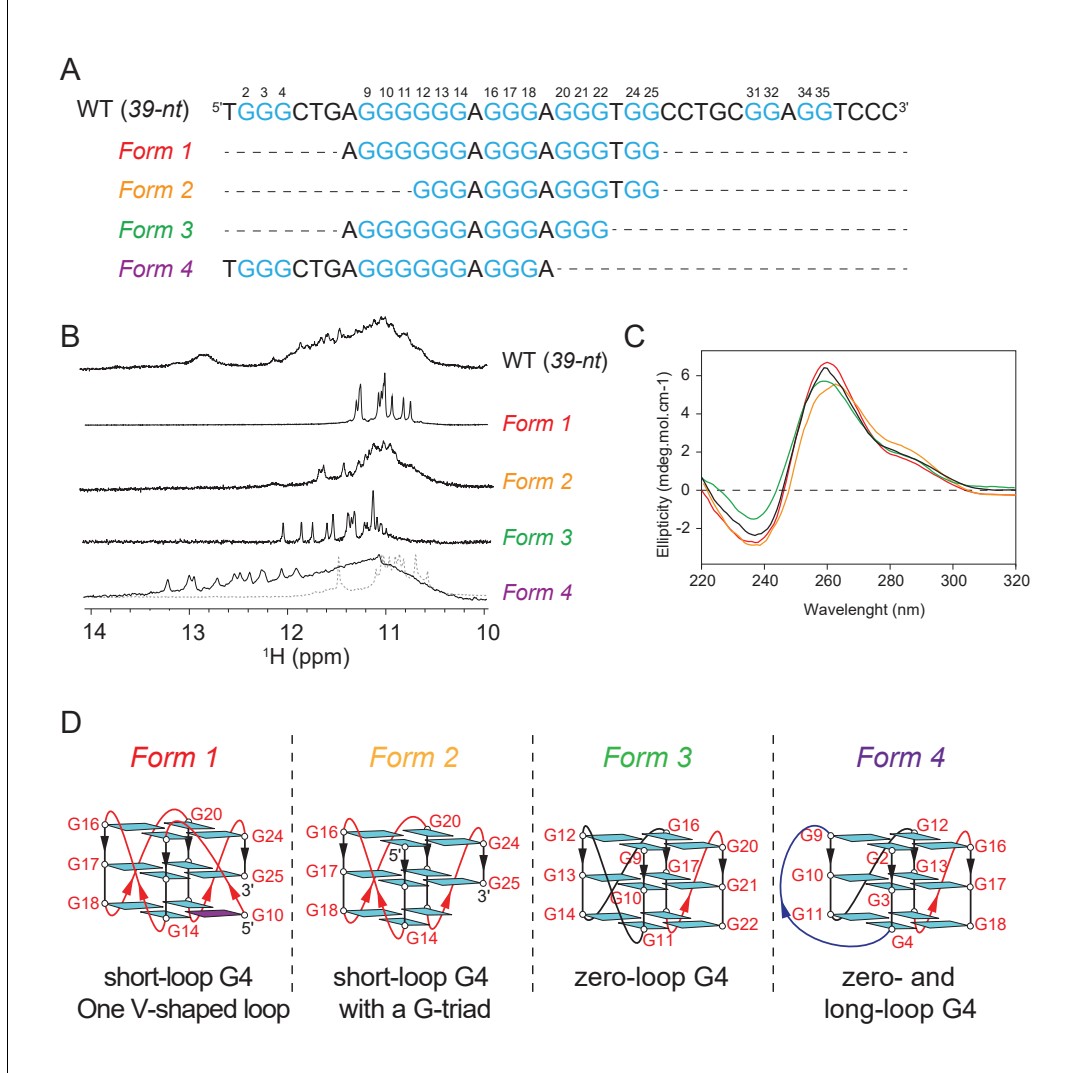

**Figure 4.** The CEB1 motif adopts multiple non-canonical G4 conformations. (**A**) Oligonucleotide sequences used to isolate individual G4 conformation in CEB1. Guanine tracts are colored cyan and numbered within a repeat unit according to their positions. (**B**) NMR spectra and (**C**) CD spectra of the mixture of G4 resulted from the CEB1-WT motif and of isolated G4 folds *Form 1, 2, 3* and *4*. The *Form 4* spectra showed in dotted line corresponds to the dimeric form observed at high $K^+$ and DNA strand concentration. The resonance frequency of the nuclei is expressed in part per million (ppM). (**D**) Isolated G4 folding topologies result from the CEB1 sequence. The snapback guanine in *Form 1* can be either $G_{10}$ or $G_{11}$, with an associated V-shaped loop containing one or zero G, respectively (*Adrian et al., 2014*). *Form 2* resembles *Form 1* but lacks the snapback guanine, and consequently exhibits a terminal G-triad. *Form 3* contains a 0-nt loop between the $G_{9-11}$ and the $G_{12-14}$ strands. The monomeric *Form 4* exhibits a 4-nt long loop between the $G_{2-4}$ and $G_{9-11}$ strands, and a 0-nt loop between the $G_{9-11}$ and $G_{12-14}$ strands. For the dimeric *Form 4* see *Figure 4—figure supplement 4J*.

The following figure supplements are available for figure 4:

**Figure supplement 1.** Multiple G4s resulted from different G-rich fragments of CEB1 minisatellite.

**Figure supplement 2.** CD spectra of the mutant CEB1 G4 motifs used in this study.

**Figure supplement 3.** NMR structural characterization of CEB1 *Form 4* in 100 mM $K^+$ solution.

**Figure supplement 4.** Imino spectral transition of *Form 4* through $K^+$ titration at ~0.1 mM strand concentration.

monomeric parallel G4 with an unusual zero-nt loop and two 1-nt propeller loops. Finally, *Form 4* can fold into a parallel G4 containing a 4-nt, a zero-nt, and a 1-nt loops. Depending on the salt and DNA strand concentration, *Form 4* is in equilibrium between an interlocked dimeric G4 and a monomeric G4 (Appendix 1, *Figure 4—figure supplements 3* and *4*). Altogether, the biophysical analyses of the highly contiguous CEB1 G-rich sub-motifs demonstrate that the CEB1 motif is able to fold into several non-canonical G4s in vitro.

## Rationalizing CEB1 instability

The mapping of the G-tracts required for each isolated G4 in vitro (*Figure 4A*) and the measure of the CEB1 variant instability in vivo (*Figures 2* and *3*) allow us to infer which G4 structures underlie the instability of the CEB1 minisatellite. While mutations such as CEB1-G(10-13)T and CEB1-G(11,13)T disfavor all identified G4 forms, in agreement with mutagenesis data showing no instability, mutations such as in CEB1-G11T, CEB1-G(3,9–11)T, CEB1-G(3,24–25)T and CEB1-G21T would result in isolation of *Form 1 + 2*, *Form 2*, *Form 3* and *Form 4*, respectively, from the other potentially competing G4s in the full motif (*Figure 5A*). Clearly, *Form 4* is not involved in CEB1 instability as the CEB1-G21T mutant remains perfectly stable in all conditions. *Form 1 + 2* accounted for 2/3rd of the

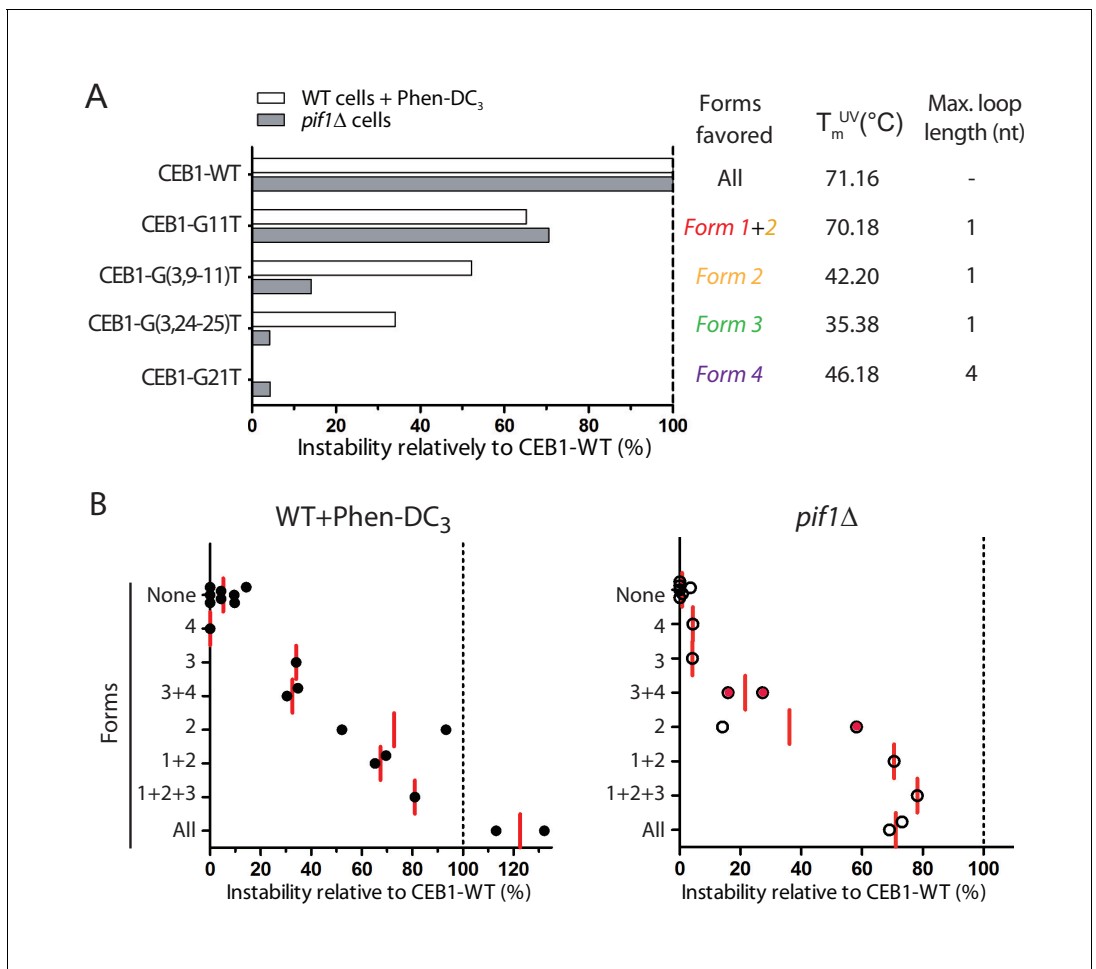

**Figure 5.** Contribution of G4 conformations to CEB1 instability. (**A**) Example of instabilities obtained in Phen-DC$_3$-treated WT cells and in *pif1Δ* cells for CEB1 variants isolating single G4 conformations, normalized to the instability obtained for CEB1-WT. Since the sequence requirements for *Form 2* is embedded in the sequence requirements for *Form 1*, no mutation could isolate *Form 1*. Its contribution can be deduced by comparing with alleles isolating *Form 2*. Thermal stability and maximum loop length for each isolated form are indicated. (**B**) Summary of the CEB1 variants instability in Phen-DC$_3$-treated WT cells and in *pif1Δ* cells as a function of the forms they can adopt. Each point corresponds to a different allele. The mean is shown in red. Red dots indicate possible formation of an unidentified form involving G$_{2-4}$.

instability in both contexts (*Figures 3D*, *5A and B*). Further isolation of *Form 2* from *Form 1* (in CEB1-G(3,9–11)T and -G(9-11)T) showed that *Form 2* is sufficient in the presence of Phen-DC$_3$ while *Form 1* preferentially causes instability in the absence of Pif1 (*Figure 5A,B*). The 0-nt containing loop *Form 3* contributed to the remaining third of instability only in the Phen-DC$_3$ context (*Figures 3D*, *5A and B*). Unfortunately, we could not isolate the putative structure(s) responsible for the remnant instability specific to the *pif1Δ* context (*Figure 5B*). A possible explanation is that the usage of the different G-triplets and the G-sextet can fold into a mixture of several poorly stable G4s due to the incorporation of various loop lengths, up to a total of 9-nts.

## Effect of G4 loops and thermal stability on CEB1 variants instability

Our previous structure-function analysis of the CEB25 minisatellite showed that short pyrimidine loop-bearing G4s were most prone to induce genomic instability, in correlation with the structure thermal stability (*Piazza et al., 2015*). Consistent with this preferential folding bias, it is remarkable that all the forms inducing CEB1 instability (*Form 1*, *2*, and *3*) bear single/zero nucleotide loops (*Figure 4A*), and that increasing a loop by a single nucleotide has such a profound effect on the array instability (*Figure 2—figure supplement 1*). To investigate the relationship between the in vivo instability of the CEB1 mutants and their in vitro thermal stability, we measured the melting temperature (T$_m$) of most CEB1 variant sequences by UV-spectroscopy in heating/cooling experiments (Materials and methods). The results reported in *Table 1* illustrate large differences in T$_m$, ranging from 71°C (CEB1-WT) to 35°C (CEB1-(G3,24–25)T). According to the G4 forms, the G11T mutant (*Form 1 + 2*) has a T$_m$ similar to CEB1-WT while all the other forms, namely *Form 2* alone (G(3,9–11)T), *Form 3* (G(3,G24-25)T) and *Form 4* (G21T) exhibit low T$_m$s (46°C or less) (*Figure 5A*). This indicates that the most prominent G4 involved in the in vivo instability has the highest T$_m$*in vitro*. More extensively, *Figure 6A* illustrates the relationship between the T$_m$ of all the variants constructed in the present study and their level of instability in the WT+Phen-DC$_3$ and *pif1Δ* contexts. Although a clear correlation can be established in both contexts, few notable outliers were observed, especially upon Phen-DC$_3$ treatment (*Figure 6A* and *Figure 6—figure supplement 1*). Among the outliers, we noted that *Form 4* did not induced instability despite a T$_m$ similar to *Form 2* and higher than *Form 3* (*Figures 5A* and *6A*). This is consistent with our previous findings that a single loop ≥4 nt was sufficient to stabilize the array independently of the T$_m$ (*Piazza et al., 2015*). We also noted that the alleles bearing isolated *Forms 2* and *3* exhibited significant instabilities in the presence of Phen-DC$_3$ but little to none in the absence of Pif1 (*Table 1* and

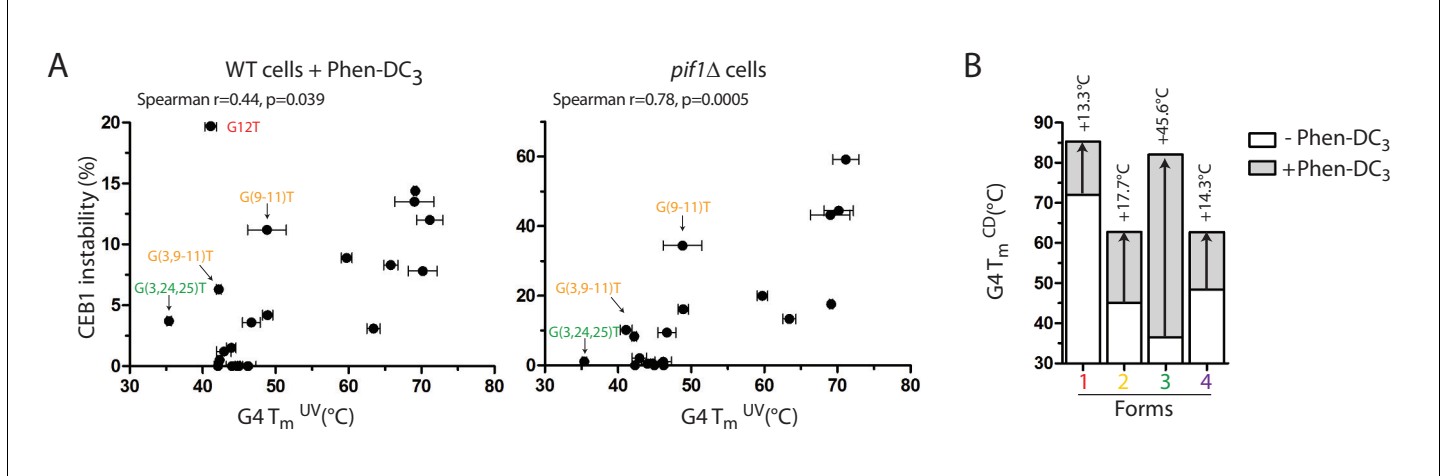

**Figure 6.** Correlation of CEB1 instability and G4 thermal stability. (**A**) CEB1 variant instabilities correlate with the thermal stabilities of their associated G4s as determined by UV-melting, in both WT cells treated with Phen-DC$_3$ (left) and in *pif1Δ* cells (right). (**B**) Phen-DC$_3$ differently stabilizes the isolated G4s resulting from CEB1, as determined by CD-melting. Arrows indicate the ΔT$_m$.
The following figure supplement is available for figure 6:

**Figure supplement 1.** Correlation between CEB1 variant instabilities upon Phen-DC$_3$ treatment and *PIF1* deletion.

*Figure 5*). To address the hypothesis that this difference results from an enhanced stabilization of these forms by Phen-DC$_3$, we compared the melting temperature of the isolated structures in the presence or absence of Phen-DC$_3$ by CD spectroscopy in heating/cooling experiments (*Figure 6B*). Indeed, we found that Phen-DC$_3$ stabilized all forms, but to varying degrees. Namely the $\Delta T_m$ for *Forms 1*, *2* and *4* were similarly increased by 13.3°C to 17.7°C but *Form 3* exhibited a large $\Delta T_m$ of +45.6°C (*Figure 6B*). The consequence of this differential effect is that *Form 1* and *3* now reach a $T_m$ of >80°C while *Form 2* and *4* have a $T_m$ hovering at ~60°C. Altogether, these results indicate that the high level of G4-induced CEB1 instability correlated with the high thermal stability of the corresponding G4 in vitro. Importantly, it revealed that Phen-DC$_3$ differentially increases the $T_m$ of the various G4s, allowing explaining certain quantitative discrepancies observed in the WT+Phen-DC$_3$ and *pif1Δ* contexts (see **Discussion**).

## Discussion

Here, we performed complementary biophysical and genetic approaches to elucidate the CEB1 G4s and study the biological consequences on genomic instability. Upon mutagenesis, we identified a core of three contiguous G-triplets (G$_{16-18}$, G$_{20-22}$, and G$_{12-14}$ within the G$_{9-14}$ sextet) as being essential for CEB1 instability. The nearby G$_{24-25}$ doublet was involved in ~2/3$^{rd}$ of the instability while the remaining instability relied on the G$_{9-11}$ triplet from the G-sextet in the Phen-DC$_3$-treated context and on the more distant G$_{2-4}$ triplet in the *pif1Δ* context (*Figure 3E*).

### Non-canonical G4s form in vivo

The various CEB1 G-tracts involved in CEB1 instability in vivo could be involved in three different G4 conformations determined in vitro: *Forms 1* to *3*. These conformations exhibited non-canonical structural features, such as a V-shaped loop and snapback guanine for *Form 1*, G-triad for *Form 2*, and zero-nt loop for *Form 3*. Hence, our structure-function analysis provides evidence for the existence of non-canonical G4s in vivo and their involvement in inducing high levels of genomic instability, as observed for the canonical CEB25-G4 motif composed of 4 G-triplets and loops of 1 nt (*Piazza et al., 2015*). Among the numerous structural studies of G4s, a 0-nt loop, such as in *Form 3*, were previously reported for the *VEGF* aptamer, although encompassing only 2 quartets (*Marušič et al., 2013*); snapback guanines occupying the vacant slot in outermost quartets such as in *Form 1* were observed in the G4 of the *c-MYC*, *c-KIT* and *PDGFRβ* promoters (*Phan et al., 2007*, *Phan et al., 2005*) and the associated V-shaped loop was reported for the *CHL1* intronic G4 (*Kuryavyi and Patel, 2010*); Finally, G-triads such as in *Form 2* have been reported for synthetic model sequences and derivatives of the human *MYOG* G4 (*Heddi et al., 2016*; *Li et al., 2015*). Interestingly, in addition to nearby guanines, the vacant slot of a triad can be filled up by freely diffusing guanine derivatives (*Park et al., 2016*). As previously proposed (*Heddi et al., 2016*; *Li et al., 2015*), G-triad-containing G4s bear the unique property of being (deoxy)riboswitches sensitive to the abundance of guanine derivatives, which act as endogenous stabilizing ligands. This adds to the list of sensors and switch functions for G4s in vivo, as proposed for temperature (*Wieland and Hartig, 2007*) and salt concentration (*Subramanian et al., 2011*).

A noteworthy observation regards the additivity for full CEB1 instability of the contributing *Forms 2* and *3* in the presence of Phen-DC$_3$, and of *Form 1* with an unidentified form in the absence of Pif1 (*Figure 5B*), suggesting the lack of interference or cooperation between forms in the array. This absence of competition can be explained if each motif has an overall low propensity to fold into a G4 at the time required to interfere with replication (*Lopes et al., 2011*), implying that G4 formation is a limiting step in CEB1 instability.

### Thermal stabilization by a G4 ligand exacerbates the effect of labile G4

By systematically determining the thermal stability of motif variants, and by varying lengths and sequences of single purine loops involved in all forms (A$_{19}$), we confirmed with the complex CEB1 sequence that loop sequence and G-tract proximity are determinants of G4-induced array genomic instability, in correlation with G4 thermal stability. This correlation was more robust in the *pif1Δ* than in the Phen-DC$_3$-treated context (*Figure 6A*), due to notable deviations affecting *Forms 2* and *3* upon Phen-DC$_3$ treatment. These differences can be explained by the disproportionate stabilization of these forms by Phen-DC$_3$ compared to other forms as previously reported for several other canonical G4s (*De Cian et al., 2007*) such as the CEB25-G4 and variants (+9 to +14°C) (*Piazza et al.,*

*2015*). Phen-DC$_3$ is a universal G4 binder due to its recognition of an exposed G-quartet (*Chung et al., 2014*). Consequently, its disproportionately high stabilization of certain non-canonical G4 likely results from the overcoming of the outmost quartet lability (due to the 0-nt loop in *Form 3* and the G-triad in *Form 2*) thanks to π-stacking interactions rather than from a differential G4 recognition. Thus, change in relative thermal stability induced by Phen-DC$_3$ explains the discrepancies observed between the Phen-DC$_3$ and the *pif1Δ* contexts for isolated forms, whereby the most stable forms induce instability in both contexts. These differences were not revealed with the canonical CEB25 motif and variants because of their monomorphic structures, all evenly stabilized by Phen-DC$_3$ (*Piazza et al., 2015*). Hence, according to the G4 motif, G4-ligands have the potential to exacerbate effects of otherwise more labile non-canonical G4s (*Chambers et al., 2015*). In our case, the discordant behavior of CEB1-G12T and CEB1-G(12,25)T, which induces more instability that CEB1-WT in the Phen-DC$_3$-treated context yet almost abolishes the instability in the Pif1-deficient context, suggests that the G12T mutation causes the formation of an uncharacterized new form that might be strongly stabilized by Phen-DC$_3$.

### Predicting G4 motifs in genomes

As outlined in the **Introduction**, a new generation of G4 prediction algorithms that takes into account non-canonical G4s re-evaluated to ~700,000 the number of potential G4 sequences in the human genome (*Bedrat et al., 2016*). The evidence for the existence of non-canonical G4s in cells revealed here gives relevance to this increased figure. However, how many of these potential G4 sequences actually form or exert a given biological function in cells remains uncertain. Indeed, a ChIP-Seq experiment using a G4-specific antibody identified only ~10,000 regions in human genomes, specifically enriched at nucleosome-depleted regions (*Hänsel-Hertsch et al., 2016*). Similarly, our studies point at only a subset of the potential G4 sequences as being 'at risk' for genomic instability. The innocuousness of other sequences could either be because G4 with longer loops did not form in vivo or because they failed to interfere with leading-strand replication (*Piazza et al., 2015*). This complex in vivo situation, affected both by the local context such as nucleosome occupancy, the presence of G4-stabilizing ligands, and likely dependent of the biological processes under scrutiny, argues against a 'one-fits-all' G4 prediction algorithm for genome data mining. With regards to genomic instability, our study identifies a subset of the most compact and stable canonical and non-canonical G4s that bears the biophysical properties required to form and hinder leading strand replication.

## Materials and methods

### Media

Synthetic complete (SC) and Yeast-Peptone-Dextrose (YPD) media have been prepared according to standard protocols (*Treco and Lundblad, 2001*). Liquid SC media containing Phen-DC$_3$ at 10 μM have been prepared as previously described (*Piazza et al., 2010*).

### Strains

Relevant genotypes of the haploid *Saccharomyces cerevisiae* strains used in this study are listed in *Supplementary file 1A*. They are derived from SY2209 (W303 *RAD5*[+] background) (*Fachinetti et al., 2010*) by Lithium-Acetate transformation (*Lopes et al., 2011*). The CEB1-WT-25 (*Ribeyre et al., 2009*), CEB1-G(9,10,14)T, CEB1-G(10-13)T, CEB1-G11T, CEB1-G(11,13)T, CEB1-G17T, CEB1-G21T, CEB1-G25T, and CEB1-G(25-24)T minisatellites have been synthesized and Sanger sequenced using a custom-made PCR-based method described previously (*Ribeyre et al., 2009*). Minisatellites of similar size (20–26 motifs) have been retained. The CEB1-G(3,9–11)T, CEB1-G(9-11)T, CEB1-G(9-11)T,G25T, CEB1-G(31-32)T, and CEB1-G(34-35)T alleles of 24 motifs have been synthesized and Sanger sequenced by GeneCust. The CEB1-WT-10 and CEB1-WT-20 (of 10 and 20 motifs, respectively), CEB1-G3T, CEB1-G(3,24–25)T, CEB1-G(9-10)T, CEB1-G(9-10,24-25)T, CEB1-G12T, CEB1-G13T, CEB1-G14T and CEB1-G(14,24–25)T alleles (24 motifs) and CEB1-A19T, CEB1-A19TT, and CEB1-A19TTT alleles (20 motifs) have been synthesized and Sanger sequenced by GenScript.

All minisatellites have been inserted at the same location and in the same orientation in the intergenic region between *YCL048w* and *YCL049c* (chrIII:41801–41840, yielding a small deletion of 39 bp) in the vicinity of *ARS305* as described previously (*Lopes et al., 2011*). Briefly, transformation of a

marker-less minisatellite fragment containing the appropriate flanking sequences replaced the *URA3-hphMX* cassette present at this location in the parental strain, allowing for the selection of the transformants (5FOA-resistant and Hygromycin-sensitive). The G-rich strand of CEB1 is on the Crick strand (e.g. template for the leading strand replication machinery of forks emanating from *ARS305*, Figure A in reference [*Lopes et al., 2011*]). CEB1-G3T contracted to 23 motifs upon insertion in the yeast genome, CEB1-G(3,24–25)T contracted to 22 motifs, CEB1-G14T, CEB1-G(9-10,24-25)T and CEB1-G(14,24–25)T all contracted to 20 motifs, CEB1-G(34-35)T contracted to 21 motifs, and CEB1-A19T contracted to 19 motifs.

## Measurement of minisatellite instability

Minisatellite instability during vegetative growth has been measured in WT, WT Phen-DC$_3$- and *pif1Δ* cells as previously described (*Ribeyre et al., 2009*) (*Lopes et al., 2011*). Briefly, untreated WT cells and *pif1Δ* cells from a fresh patch of cells are diluted at a concentration of $2 \times 10^5$ cells/mL in 5 mL of YPD, grown at 30°C with shacking for eight generations, spread as single colonies on YPD plates, and incubated at 30°C. The instability measurement in these cells thus corresponds to the rearrangement frequency after 35 generations. Between 48 and 192 colonies from these patches were analyzed (see below for sample size determination) in a single experiment. In rare instances in which an early clonal 'jackpot' event was present in the starting colony, we analyzed an independent patch. To measure minisatellite instability upon Phen-DC$_3$ treatment, cells from a fresh patch were grown for 8 generations at 30°C in liquid SC containing 10 μM Phen-DC$_3$ (*Lopes et al., 2011*). Isolated colonies or pools of colonies are analyzed by Southern blot upon digestion with *Eco*RI that cut at each side of the minisatellite, leaving a total of 18 nt of flanking sequence. The membranes are hybridized with the Phage lambda DNA (*Hind*III/*Eco*RI digested ladder, Promega) and the appropriate CEB1-WT or variant probes. The signals are detected with a Typhoon Phosphorimager and quantified using ImageQuant 5.2 (Molecular Dynamics). The elimination of secondary rearrangements (that occurred early in the colony after plating) and of potential early clonal events in the culture has been performed as described in (*Lopes et al., 2011*). We used G*Power to compute sample size, with a α cutoff set at 0.05. Given the usual range of CEB1 instability observed in untreated and Phen-DC$_3$ treated WT cells (based on previous studies), and to be able to detect instabilities of at least 5% with an α cutoff of 0.05 and a β power of 0.9, we set the sample size at 192 colonies. This sample size also allows detecting a 2-fold decrease of instability compared to the reference CEB1-WT allele in the Phen-DC$_3$ context. Regarding the CEB1 instability in *PIF1*-deleted cells, we knew from previous study that the range of instabilities was much wider, reaching very high levels (up to 60% for CEB1-WT). These high instabilities prevented colony pooling for Southern blot analysis. The level of instability was hinted at upon verification of the transformants by Southern blot. Hence, when the instability was expected in the high range, we chose to sample 48 colonies, a reasonable compromise which allows detecting instability levels of 15% compared to WT cells, and instabilities 2-fold lower than in CEB1-WT with a β power of 0.9. For predicted intermediate instabilities we analyzed 96 colonies, or otherwise 192 colonies as in the WT context.

## DNA sample preparation

Unlabeled and site-specific labeled DNA oligonucleotides (*Supplementary file 1B*) were chemically synthesized on an ABI 394 DNA/RNA synthesizer. Samples were purified and dialyzed successively against 25 mM potassium chloride solution and water. Unless otherwise stated, DNA oligonucleotides were dissolved in solution containing 70 mM potassium chloride and 20 mM potassium phosphate (pH 7.0). DNA concentration was expressed in strand molarity using a nearest-neighbor approximation for the absorption coefficients of the unfolded species (*Cantor et al., 1970*).

## Gel electrophoresis

The molecular size of the structures formed by DNA oligonucleotides was visualized by non-denaturing polyacrylamide gel electrophoresis (PAGE) (*Guédin et al., 2008*). DNA samples were incubated in a 20 mM potassium phosphate buffer (pH 7.0) before loading on 20% polyacrylamide gels supplemented with variable concentration of potassium chloride and run at 26°C; 40% sucrose was added before loading.

## Circular dichroism

Circular dichroism (CD) spectra were recorded on a JASCO-810 spectropolarimeter using 1 cm path length quartz cuvettes with a reaction volume of 600 µL. The DNA oligonucleotides (~5 µM) were prepared in a 20 mM potassium phosphate buffer (pH 7.0) containing 70 mM potassium chloride. For each experiment, an average of three scans was taken, the spectrum of the buffer was subtracted, and the data were zero-corrected at 320 nm.

## Thermal difference spectra

The thermal difference spectra (TDS) were obtained by taking the difference between the absorbance spectra of unfolded and folded oligonucleotides that were respectively recorded much above and below its melting temperature. TDS provide specific signatures of different DNA structural conformations (*Mergny et al., 2005*). Spectra were recorded between 220 and 320 nm on a JASCO V-650 UV/Vis spectrophotometer using 1 cm pathlength quartz cuvettes. The DNA oligonucleotides (~5 µM) were prepared in a 20 mM potassium phosphate buffer (pH 7.0) containing 70 mM potassium chloride. For each experiment, an average of three scans weve taken, and the data were zero-corrected at 320 nm.

## Circular dichroism and UV melting experiments

The thermal denaturing of the CEB1-WT and its mutants was performed on JASCO UV/VIS V-650 spectrophotometer or on a CD by monitoring the UV absorption (at 290 nm wavelength) or the CD ellipticity (at 260 nm wavelength). Prior to melting experiments, DNA samples (5–10 µM) were annealed in a buffer containing 10 mM potassium chloride and 10 mM potassium phosphate (pH 7). All melting experiments were performed using the protocols described in (*Piazza et al., 2015*). Melting experiments with Phen-DC$_3$ were conducted at DNA:Phen-DC$_3$ ratio of 1:1.

## NMR spectroscopy

NMR experiments were performed on 600 MHz and 700MHz Bruker spectrometers at 25°C, unless otherwise specified. The strand concentration of the NMR samples was typically 0.2–1.5 mM in near-physiological conditions (100 mM K$^+$ solution at pH 7). Resonances for guanine residues were assigned unambiguously by using site-specific low-enrichment $^{15}$N labeling (*Phan and Patel, 2002*), site-specific $^2$H labeling (*Huang et al., 1997*), and through-bond correlations at natural abundance (*Phan, 2000*). Spectral assignments were completed by NOESY, TOCSY, {$^{13}$C-$^1$H}-HMBC and {$^{13}$C-$^1$H}-HSQC as previously described (*Phan et al., 2001*). Inter-proton distances were deduced from NOESY experiments at various mixing times. All spectral analyses were performed using the FELIX (Felix NMR, Inc.) program.

## Statistical analysis

Sample size determination for instability measurement are described in the 'Measurement of minisatellite instability' section. The rearrangement frequencies have been compared using a two-tailed Fisher exact test using R x64 3.2.0 (*Team, 2008*). A non-parametric Spearman correlation test has been used to compare thermal stability of the G4 variants and the associated CEB1 allele instabilities. In all cases, the α-cutoff for significance has been set to 0.05.

## Acknowledgements

We thank the members of our laboratory as well as Maria Emilia Puig Lombardi and A Londono-Vallejo for helpful discussions. We also thanks Marie-Paule Teulade-Fichou for the Phen-DC$_3$ compound and helpful discussions. We are grateful to Allyson Holmes for her careful reading and editing of the manuscript. This research was supported by the Agence Nationale de la Recherche ANR-12-BSV6-0002-01 and ANR-14-CE35-0003-02 (to AN), Singapore Ministry of Education Academic Research Fund Tier 3 (MOE2012-T3-1-001) and Nanyang Technological University grants (to ATP) and the Singapore-France Merlion grant (to ATP and AN). AP was supported by graduate fellowships from the MNERT and the ARC. MA was supported by the Yousef Jameel scholarship. XC was supported by a postdoctoral fellowship from the Institut Curie, Research Center and the ANR-14-CE35-0003-02.

## Additional information

### Funding

| Funder | Grant reference number | Author |
| --- | --- | --- |
| Agence Nationale de la Recherche | ANR-12-BSV6-0002-01 | Alain G Nicolas |
| Ministry of Education - Singapore | MOE2012-T3-1-001 | Anh-Tuan Phan |
| Nanyang Technological University | | Anh-Tuan Phan |
| Fondation ARC pour la Recherche sur le Cancer | Doctoral fellowship | Aurele Piazza |
| Agence Nationale de la Recherche | ANR-14-CE35-0003-01 | Alain G Nicolas |
| MENRT Fellowship | Doctoral fellowship | Aurele Piazza |

The funders had no role in study design, data collection and interpretation, or the decision to submit the work for publication.

### Author contributions

AP, XC, Conceived the project, Designed, performed and analyzed the in vivo experiments, Wrote the manuscript; MA, Conceived the project, Designed, performed and analyzed the in vitro experiments, Wrote the manuscript; FS, Designed, performed and analyzed the in vivo experiments; BH, Designed, performed and analyzed the in vitro experiments, Wrote the manuscript; A-TP, AGN, Conceived the project, Wrote the manuscript

### Author ORCIDs

Aurèle Piazza, http://orcid.org/0000-0002-7722-0955
Xiaojie Cui, http://orcid.org/0000-0003-0711-8178
Frédéric Samazan, http://orcid.org/0000-0002-4175-2810
Brahim Heddi, http://orcid.org/0000-0002-5535-6237
Anh-Tuan Phan, http://orcid.org/0000-0002-4970-3861
Alain G Nicolas, http://orcid.org/0000-0002-5606-7808

## Additional files

### Supplementary files

• Supplementary file 1. Strains and primers used in this study. (A) Relevant genotypes of haploid W303 *Saccharomyces cerevisiae* strains used in this study. (B) List of CEB1 sequences analyzed using NMR spectroscopy in vitro.

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

# Appendix 1: CEB1 *Form 4*: Equilibrium between an interlocked dimeric G4 and a monomeric G4 containing a zero-nt loop

The imino spectrum of *Form 4* exhibited twelve distinguished peaks at resonance region (10-12 ppm) indicative of a well-defined G4 formed by the 19-nt sequence (*Figure 4—figure supplement 3A*) (*Adrian et al., 2012*). Thermal difference spectra (TDS) of *Form 4* also displayed a G4 pattern with two positive maxima at 240 and 275 nm and a negative minimum at 295 nm (*Figure 4—figure supplement 3B*) (*Mergny et al., 2005*). CD spectrum of *Form 4* showed a positive maximum at 260 nm and a negative minimum at 240 nm, which is a signature of parallel structure (*Figure 4—figure supplement 3C*) (*Gray et al., 2008*).

Guanine imino (H1) and aromatic (H8) protons of *Form 4* were unambiguously assigned using site-specific low-enrichment $^{15}$N labeling (*Phan and Patel, 2002*), site-specific substitution $^{2}$H labeling (*Huang et al., 1997*), and through-bond correlations at natural abundance ($\{^{13}$C-$^{1}$H$\}$-HMBC) (*Figure 4—figure supplement 3D,E*) (*Phan, 2000*). The unambiguous assignments were used to follow the H8/H6-H1' NOE sequential connectivity from T1 to A19 and to identify cross-peaks on NOESY spectra (*Figure 4—figure supplement 3F,G,H and I*). Discontinued NOE connections before and after A15 is the characteristic pattern of a double-chain-reversal loop with A15 being the loop residue (*Figure 4—figure supplement 3J*). All intra-residue H8-H1' NOEs were of medium intensity, indicating anti glycosidic conformation for all tetrad-bound guanines.

The observed H1-H8 NOE cyclic patterns around G-tetrads established the folding of an interlocked dimeric G4 encompassing six G-tetrads that is, G2•G9•G2*•G9*, G3•G10•G3*•G10*, G4•G11•G4*•G11*, G12•G16•G12*•G16*, G13•G17•G13*•G17* and G14•G18•G14*•G18* (residues from the second strand are marked with asterisks) (*Figure 4—figure supplement 3H,I and J*). The structure is symmetrically arranged in parallel and supported by two main G4 backbones (G9-G10-G11-G12-G13-G14) uninterruptedly stretching from 5'-side to 3'-side of the molecule. The primary backbone joins G2-G3-G4 and G16-G17-G18 columns by a 4-nt (C5-T6-G7-A8) and a 1-nt (A15) double-chain-reversal loop, respectively. Viewed from 5'-end of the strand, the hydrogen bond directionalities of G-tetrad are all clockwise.

Real time proton exchange experiment was done by dissolving the sample in D$_2$O solution. Eight imino peaks (i.e., from G3, G4, G10, G11, G12, G13, G17 and G18) remained for longer than 3 weeks at 25 °C, while the other four imino peaks (G2, G9, G14 and G18) disappeared after 15 minutes exposure at the same condition (Figure Supplement 3E). The position of G3•G10•G3*•G10*, G4•G11•G4*•G11*, G12•G16•G12*•G16* and G13•G17•G13*•G17* tetrads at the middle layers results in protection of guanine imino protons from exchanging with solvent protons, in agreement with the observation of prolonged imino proton lifetime of the corresponding guanines (*Figure 4—figure supplement 3J*).

The sequence of *Form 4* produced non-resembling imino proton peak patterns depending on both strand and cation concentrations, buffer condition and sample preparation, therefore suggesting the presence of different species of *Form 4* (*Figure 4—figure supplement 3K*). Additionally, gradual spectral transition between *Form 4* species could be observed by the increment of K$^+$ concentration (*Figure 4—figure supplement 4*). A parallel stranded G4 fold of monomeric *Form 4* was deduced based on the following: (i) lower molecular weight of *Form 4* species (resulted from dissection of pre-folded dimeric *Form 4*) visualized by non-denaturing PAGE experiments (*Figure 4—figure supplement 3L*), (ii) CD data obtained in 5 mM K$^+$ solution at ~4 µM strand concentration (*Figure 4—figure supplement 3M*), and (iii) *Form 4* sequence constraint. The proposed model G4 comprises G2•G9•G12•G16, G3•G10•G13•G17 and G4•G11•G14•G18 tetrads and three propeller loops

including 4-nt and 1-nt side loops and a rarely observed zero-nucleotide central loop (*Figure 4—figure supplement 3N*).

