## [Decision Letter]

Thank you for submitting your article "Non-Canonical G-quadruplexes Cause the hCEB1 Minisatellite Instability in *Saccharomyces cerevisiae*" for consideration by *eLife*. Your article has been reviewed by three peer reviewers, one of whom is a member of our Board of Reviewing Editors, and the evaluation has been overseen by Diethard Tautz as the Senior Editor. The reviewers have opted to remain anonymous.

The reviewers have discussed the reviews with one another and the Reviewing Editor has drafted this decision to help you prepare a revised submission.

Summary:

This is a carefully executed study that identifies novel G4 non-canonical motifs crucial for the induction of genetic instability. The authors report an extensive structure-function analysis of multiple arrays of the human CEB1 G4-forming sequence. This motif has a complex composition: G3-N4-G6-A-G3-A-G3-T-G2-N5-G2-A-G2. Consequently, formation of multiple G4 configurations was possible and the authors generated a set of 27 mutant hCEB1 motifs containing changes in G runs and sizes of the loops. For the in vivo assessment of genomic instability associated with the G4 forming sequence, wild-type or mutated CEB1 variants arrays were inserted close to ARS305 in *S. cerevisiae* (in the orientation where the G-rich strand is template for leading strand synthesis) and the number of rearrangement counted in untreated wild-type cells, wt cells treated with G4-stabilizing Phen_DC3 ligand and in pif1∆ cells. These in vivo data were correlated with in vitro analysis of mutant motifs using NMR, CD and thermal stability spectra. The authors' biophysical studies in vitro convincingly showed formation of four dominant G4 structures, none of which meets the canonical requirements for GV4-formation: they contain 1 or 0 nt-long loops, snap-back guanines, G-triads instead of G-quartets, etc. This combined analysis allowed to identify non-canonical G4s inside hCEB1 and define an impact of different G4 conformations on the minisatellite instability. Among other conclusions it is worth mentioning the presence of 2 central G-triplets and a short distance between them are crucial for the destabilization of hCEB1, that the G-sextet is another run required for the destabilization whereas only 3Gs in the sextet separated by 1 nt from the 2 central G-triplets contribute to instability, that the destabilizing role of distantly located G4 triplet is only seen when the tract is missing G2-doublet and only in the pif1 mutants or the identification of 3 overlapping motifs required for instability (G3AG3AG3TG2, Phen-DC3-dependent G6AG3AG3 and pif1-dependent G3N4G6AG3AG3) that are non-canonical deviations from G3N1-7G3N1-7G3N1-7G3 consensus. These observations point to the fact that G4-structures are even more versatile than it was believed before. The results strongly support the author's conclusions, and the manuscript is well-written. Importantly, this work demonstrates that currently used G4 prediction algorithms are not perfect and experimental work is required to determine the real destabilizing potential of putative G4 motifs.

Essential revisions:

The manuscript is difficult to follow because it is not written for a broad audience. It needs to be heavily worked out to make it much clearer for readers of a general journal like *eLife*, and not for a specialized journal. It is also important to explain why the deviations from the canonical G4-structure/motif matter at all.

Regarding the results described in the subsection “CEB1 instability relies on two G-triplets and short loop length” paragraph two, it is not clear to what wt the data is compared to (according to the numbers, is seems it is compared to WT-25 in Phen-DC3-treated cells and to WT-20 in pif1∆). It would also help the understanding if these controls were depicted in Figure 1 and not only in the table. Similar comment on results described in the subsection “CEB1 instability largely relies on a G-doublet”: which CEB1-WT is it compared to?

Discussion on 'non-canonical G4s form in vivo': Conclusions are based on correlations between in vitro structure analysis and in vivo phenotype and not on in vivo structure analyses. This should be better explained.

A correlation between Tm in vitro and the capacity to destabilize the genome in vivo seems clear. Therefore, it should be possible to play with the temperature used in vivo to analyze instability. In principle culturing cells at higher or lower temperature should have a strong impact on the stability of some of the constructs, in particular those with the highest Tm. Some of the constructs with the highest instability should be used for this.

The table before supplemental needs a better explanation. It seems to be a summary of all results, and suggest a number of questions. For example, the data indicated that there are motifs highly stable with Phe-DC3 and less stable in pif1∆, and vice-versa. A detailed attention to those mutants and possible differential mechanism of stability should be given, whether or not with further experiments that try to explain the different behaviours of these constructs in vivo.

---

## [Author Response]

*Essential revisions:*

*The manuscript is difficult to follow because it is not written for a broad audience. It needs to be heavily worked out to make it much clearer for readers of a general journal like eLife, and not for a specialized journal. It is also important to explain why the deviations from the canonical G4-structure/motif matter at all.*

Our constant goal in writing this manuscript has been to present the (sometimes complex) results and interpretation in its most accessible fashion, by avoiding jargon and being as concise as possible. Still, it became difficult to avoid an irreducible body of descriptive language, well used in the G4 and biophysics field. In order not to antagonize the broad readership of *eLife*, we made three significant changes:

We added an introductory figure that visually provides first level background information on: (Figure 1) the G-quadruplex structural features and descriptive terminology, (Figure 1) the experimental conditions in which G4 forms and are stabilized by the ligand or in the absence of the pif1 helicase and (Figure 1) show example of Southern blots allowing to measure CEB1 instability. In parallel, additional information was added in the text and in the Methods section, in order to spare the reader to find our previous publications. This should help non-specialized readers to immediately grasp the rationale of the assay and potential differences between conditions.

We removed the description of structural polymorphisms (such as V-shaped loops, G-triad, etc.) from the Introduction, and maintained it to a minimum in the structural part of the Results and Discussion. For the general audience, understanding that the CEB1 sequence form polymorphic and non-canonical G4 structures is already getting two strong points of the results. Considering that the detailed nature of the polymorphisms is mostly of interest for the specialists, we are comfortable to have moved it later in the manuscript.

We also added, in the Introduction, a statement as for why knowing which sequences form G4 in cells (being canonical or not) matters (first paragraph).

*Regarding the results described in the subsection “CEB1 instability relies on two G-triplets and short loop length” paragraph two, it is not clear to what wt the data is compared to (according to the numbers, is seems it is compared to WT-25 in Phen-DC3-treated cells and to WT-20 in pif1∆). It would also help the understanding if these controls were depicted in Figure 1 and not only in the table. Similar comment on results described in the subsection “CEB1 instability largely relies on a G-doublet”: which CEB1-WT is it compared to?*

We thank the reviewer for noticing this mistake: The CEB1-A19T should indeed be compared to CEB1-WT-20 in both cases.

We now include the blots for CEB1-WT-25 as examples in the introductory Figure 1. They were already published in our previous publication (Piazza et al., 2015). This is made clear in the Results, subsection “Experimental system” and in Figure 1 legend.

Regarding subsection “CEB1 instability largely relies on a G-doublet”, we presented the frequencies for CEB1-G24, 25T and CEB1-G25T without comparing to CEB1-WT (in this case WT-25). We now include this comparison for clarity.

*Discussion on 'non-canonical G4s form in vivo': Conclusions are based on correlations between in vitro structure analysis and in vivo phenotype and not on in vivo structure analyses. This should be better explained.*

Indeed, we infer the in vivo structure from the in vitroexperiment. To clarify this, we now start the Results, subsection “Rationalizing CEB1 instability” as follows:

“The mapping of the G-tracts required for each isolated G4 in vitro(Figure 4) and the measure of the CEB1 variant instability in vivo (Figure 2 and Figure 3) allow us to infer which G4 structures underlie the instability of the CEB1 minisatellite”.

*A correlation between Tm in vitro and the capacity to destabilize the genome in vivo seems clear. Therefore, it should be possible to play with the temperature used in vivo to analyze instability. In principle culturing cells at higher or lower temperature should have a strong impact on the stability of some of the constructs, in particular those with the highest Tm. Some of the constructs with the highest instability should be used for this.*

We also previously considered testing the instability of a minisatellite at different temperatures, but never convinced ourselves that we might learn something interpretable for several reasons. First, the lack of knowledge on the effect of the temperature on the stability and metabolism of the Phen-DC_3_ ligand; second, the limited temperature range that can be realistically assayed in cells, which might not be sufficient to reveal small quantitative variations. Finally, and foremost, growth temperature has pleiotropic effects on cell physiology, including significant changes in gene expression, that even if some reproducible effects are observed will not allow to firmly conclude on a direct variation in G-quadruplex stability. Considering these caveats and the fact that it would not bolster the main conclusions of this study (non-canonical G4 form and cause genomic instability in cells), we prefer not to engage in this experimental path and substantially delay publication.

*The table before supplemental needs a better explanation. It seems to be a summary of all results, and suggest a number of questions. For example, the data indicated that there are motifs highly stable with Phe-DC3 and less stable in pif1∆, and vice-versa. A detailed attention to those mutants and possible differential mechanism of stability should be given, whether or not with further experiments that try to explain the different behaviours of these constructs in vivo.*

This is main Table 1, which we extensively refer to throughout the text. We are sorry that it did not appear properly labeled in the submitted manuscript. We corrected it in the revised version (Table 1,Genomic instabilities of CEB1 variants and the associated G4 thermal stability.)

Regarding the differential behaviors of certain CEB1 variants in Phen-DC_3_ versus *pif1△*: we were indeed very interested in these results as they were the first instances of discrepancies between the two assay conditions in our literature on the topic (Piazza et al., 2010, Lopes et al., 2011, Piazza et al., 2012 and Piazza et al., 2015). We first point this out in the Results section. We did investigate the underlying reasons in detail and concluded that the disproportionate stabilization by Phen-DC_3_ of *Form 3* (and to a lesser extent *Form 2*) compared to *Form 1* and *4* explains most of these discrepancies (please refer to Results, subsection “Effect of G4 loops and thermal stability on CEB1 variants instability”, Figure 6 and the Discussion section dedicated to these observations, subsection “Thermal stabilization by a G4 ligand exacerbates the effect of labile G4”). The unique case of CEB1-G12T that we did not further investigated is discussed.